

# A Hydrological Cycle Model for the Globally Resolved Energy Balance Model (GREB) v1.0

Christian Stassen[1], Dietmar Dommenget[1], Nicholas Loveday[2]

[1]School of Earth Atmosphere and Environment, ARC Centre of Excellence for Climate System Science, Monash University, Melbourne, Australia
[2]Bureau of Meteorology, Darwin, Northern Territory, Australia

*Correspondence to*: Christian Stassen (christian.stassen@monash.edu)

**Abstract.** In this study, we describe the development of the hydrological cycle for the Globally Resolved Energy Balance (GREB) model. Starting from a very simple zero order hydrological cycle model included in the GREB model, we develop three new models: precipitation, evaporation and horizontal transport of water vapour. Precipitation is modelled based on the actual simulated specific and relative humidity in GREB and the prescribed boundary condition of vertical velocity. The evaporation bulk formula is slightly refined by considering differences in the sensitivity to winds between land and oceans, and by improving the estimates of the wind magnitudes. Horizontal transport of water vapour is improved by approximating moisture convergence by vertical velocity. The new parameterisations are fitted against observations (GPCP) and reanalysis data sets (ERA-Interim). The new hydrological cycle is evaluated against CMIP model simulation, reduction in correction terms and by three different sensitivity experiments (Annual Cycle, El Nino Southern Oscillation and Climate Change). The skill of the hydrological cycle in the GREB model is now within the range of more complex CMIP5 CGCMs and capable of simulating key features of the climate system within the range of uncertainty of CMIP5 model simulations. The results illustrate that the new GREB model's hydrological cycle is a useful model to study the climate's hydrological response to external forcings and also to study inter-model differences or biases.

## 1 Introduction

One topic in climate change that deserves urgent attention is the changing pattern of the hydrological cycle and of rainfall (Donat et al., 2016). Changes of rainfall have direct impact on the environment and on human health (Dai, 2011; Parry et al., 2004; Patz et al., 2005). The projections on how rainfall is changing are primarily based on General Circulation Models (CGCMs). CGCMs evaluated by the Intergovernmental Panel on Climate Change (IPCC) for the fourth assessment report (AR4), are the best possible and most complex simulations of the climate system. But, it's far from trivial to understand even simple aspects of the climate system because several processes interact with each other (Dommenget & Floter, 2011).

Rainfall is generated by a multitude of different systems (e.g. mid-latitude cyclones, tropical convection), which makes it one of the most complex processes in the climate system to model and thus to forecast. Yet many aspects of the hydrological cycle (i.e. high precipitation in the ITCZ) seen in complex CGCMs can be found in models with intermediate complexity such as the 'CLIMBER-2' (Petoukhov et al., 1999), the 'UVic earth system climate model' (Weaver et al., 2001) or the



simple atmosphere-ocean-sea-ice model developed by Wang and Myask (2000). Additionally, idealized models like the omega and humidity based model by Pendergrass and Gerber (2016) or the simple enhanced advection model by Chadwick et al. (2016) are capable of representing many aspects of the climate change response seen in complex CGCMs. Simplified climate models and energy balance considerations, are capable of explaining the large-scale features of the climate system

and climate change (e.g. arctic amplification and land-sea contrast (Dommenget & Floter, 2011; Izumi et al., 2015). They provide a framework to conceptually understand the hydrological response to climate change. Because of their simplicity, they help to develop hypotheses about the processes involved.

In this paper, we present a simple hydrological cycle model for the Globally Resolved Energy Balance (GREB) model (Dommenget & Floter, 2011). The aim of this hydrological cycle model is to present a simple and fast model for studies of

the large-scale climate in precipitation, its response to climate variability (e.g. El Nino or climate change) and external forcings. The GREB model was originally developed to simulate the surface temperature globally and in particular its response to a $CO_2$ forcing. The hydrological cycle in the GREB model was only needed as a zero order estimate to model the latent heat in the energy balance and the atmospheric water vapour levels. We now aim for a representation of the hydrological cycle, that will allow to study the hydrological cycle in the GREB model by itself. We will improve three

separate parameterisations: precipitation, evaporation and the circulation of water vapour in the model. The model will be based on the dynamical variables (surface and atmospheric temperature and humidity) in the GREB model and on the boundary conditions of the GREB model (horizontal and vertical winds).

This paper is organised as follows, in the following section, we present the data sets used, the original GREB model and the methods. In Section 3 the new parameterisations of the hydrological cycle in the GREB model is described. In the Model

Validation Section 4, we present three different sensitivity experiments to test the new hydrological cycle model. Finally, we give a discussion and summary of the results.

## 2 Data and Methods

The original GREB model used climatological fields from the NCEP reanalysis data from 1950 to 2008 (Kalnay et al., 1996) for surface temperature, $T_{surf}$, specific humidity and horizontal winds. The cloud climatology is taken from the ISCCP project

(Rossow & Schiffer, 1991). The ocean mixed layer depth is taken from Lorbacher et al. (2006). Topographic data is taken from the ECHAM5 atmosphere model (Roeckner et al., 2003). For more details refer to Dommenget and Floter (2011). For development of the new GREB hydrological cycle model we replaced the NCEP reanalysis boundary conditions for $T_{surf}$, specific humidity and horizontal winds by using ERA-Interim reanalysis data from 1979 to 2015 (Dee et al., 2011). The reasoning for this change data sets is further explained in section 3.4. Precipitation observations are taken from the Global

Precipitation Climatology Project (GPCP) (Adler et al., 2003). The climatological boundary conditions and constraints for the GREB model are summarised in Figure 1. In the following we will refer to these datasets as observations.



The observed hydrological cycle in terms of the annual mean and its seasonal cycle for precipitation, evaporation and moisture circulation are shown in Figs. 2 and 3. The global pattern of precipitation is marked by the inner tropical convergence zone (ITCZ) and its seasonal cycle, and by the storm tracks of the midlatitudes. The evaporation is strongest over subtropical oceans and has a complex seasonal cycle with in general more evaporation in the warm season over land.

The horizontal moisture transport (Figure 2c and 3c) is dominated by large scale convergence and divergence zones over the oceans and their seasonal shift.

Model simulations, pre-industrial (pi-Control) and representative concentration pathway 8.5 (rcp85), from the Coupled Model Inter-comparison Project phase 5 (CMIP5) database are used for comparison (Taylor et al., 2012). All datasets are re-gridded to a horizontal resolution of 3.75 x 3.75 to match the GREB grid. See **Table 1** for a complete list of models used.

The GREB model is a three layer (surface, atmosphere and deep ocean) global climate model on a 3.75 x 3.75 horizontal latitude-longitude grid. GREB simulates the thermal (long-wave) and solar (short-wave) radiation, heat transport in the atmosphere by isotropic diffusion and advection with the mean winds, the hydrological cycle (evaporation, precipitation and water vapour transport), a simple ice/snow albedo feedback and heat uptake in the sub-surface ocean. The model main time step is 12hrs and the daily cycle of incoming solar radiation is not resolved (e.g. 24hrs mean incoming solar radiation). The

tendency equation of surface temperature, deep ocean temperature and specific humidity are flux corrected towards reanalysis data. The wind and cloud cover field are seasonally prescribed boundary conditions. Thus, the GREB model is conceptually very different from the CGCM simulations in CMIP5, as atmospheric and the oceanic circulations are not simulated. Additionally, the GREB model has no internal variability, as daily weather systems are not simulated. Subsequently, the control climate or response to forcings can be estimated from one single year.

The original GREB hydrological cycle model, which is the starting point for this study, is shortly presented below. All variables and parameters are listed and explained in Table 2. The precipitation is proportional to the specific humidity

$$\Delta q_{precip} = r_{precip} \cdot q_{air} \qquad (1)$$

with Eq. (1), which corresponds to a autoregressive model with a decorrelation (recirculation) time of about 14 days (Dommenget & Floter, 2011). Evaporation, Eq. (2), in the original GREB model is calculated using an extended bulk

formula:

$$\Delta q_{eva} = \frac{\rho_{air} \cdot c_w \cdot |u_* + c_{turb}| \cdot \vartheta_{soil} \cdot (q_{air} - q_{sat})}{r_{qviwv}} \qquad (2)$$

The Bulk formula depends on the saturation deficit ($q_{air} - q_{sat}$), the wind speed $u_*$, with a turbulent wind factor $c_{turb}$, the density of air $\rho_{air}$, the transfer coefficient $c_w$, and a linear regression factor, $r_{qviwv}$, which links surface humidity to the vertically integrated water vapour column (Dommenget & Floter, 2011; Rapti, 2005).

The saturation water vapour pressure is calculated after (Dommenget & Floter, 2011; James, 1995):

$$q_{sat} = e^{\frac{z_{topo}}{z_{atmos}}} \cdot 3.75 \cdot 10^{-3} \cdot e^{17.08085 \frac{T_{surf} - 273.15}{T_{surf} - 38.975}} \qquad (3)$$

Together, this leads to the complete tendency equation of specific humidity in GREB



$$\frac{dq_{air}}{dt} = \Delta q_{eva} + \Delta q_{precip} + \kappa \cdot \nabla^2 q_{air} - \vec{u} \cdot \nabla q_{air} + \Delta q_{correct} \tag{4}$$

with the diffusion term $\kappa \cdot \nabla^2 q_{air}$, the advection term $\vec{u} \cdot \nabla q_{air}$ and the flux correction term $\Delta q_{correct}$. The simulated annual mean and seasonal cycle for precipitation, evaporation and mean horizontal moisture transport are shown in Figure 2 and 3 for the original GREB model as discussed above. The original GREB model is simulating some of the main features of the regional differences in the precipitation and evaporation, but many important details are missing (e.g. ITCZ). However, horizontal moisture transport is not simulated well by the original GREB model.

The seasonally varying correction term is calculated as the residuum between the tendencies without flux corrections and observed tendencies:

$$\Delta q_{correct} = \Delta q_{eva} + \Delta q_{precip} + \kappa \cdot \nabla^2 q_{air} - \vec{u} \cdot \nabla q_{air} - \left. \frac{dq_{air}}{dt} \right|_{obs} \tag{5}$$

This effectively corrects the GREB model to have a climatological specific humidity as observed. The flux correction term $\Delta q_{correct}$ can help to evaluate the improvements in the hydrological cycle model. The better the model the smaller the correction term should be in Eq. (4). We can therefore split the flux correction into three diagnostic terms

$$\Delta q_{correct} = \Delta q_{cor-precip} + \Delta q_{cor-evapo} + \Delta q_{cor-circul} \tag{6}$$

With each term on the RHS representing the fraction of the flux corrections attributed to precipitation, evaporation and circulation biases, respectively. Each term is estimated as the difference between the observed and the GREB model tendencies of the humidity resulting from precipitation, evaporation and circulation biases:

$$\Delta q_{cor-precip} = \Delta q_{precip-OBS} - \Delta q_{precip-GREB} \tag{7}$$

$$\Delta q_{cor-evapo} = \Delta q_{evapo-OBS} - \Delta q_{evapo-GREB} \tag{8}$$

$$\Delta q_{cor-circul} = \Delta q_{circul-OBS} - \Delta q_{circul-GREB} \tag{9}$$

with the GREB model tendencies of the humidity resulting from circulation, $\Delta q_{circul-GREB}$, defined as:

$$\Delta q_{circul} = \kappa \cdot \nabla^2 q_{air} - \vec{u} \cdot \nabla q_{air} \tag{10}$$

The observed humidity tendencies resulting from circulation, $\Delta q_{circul-GREB}$, are defined by the residual of the total humidity tendency minus the precipitation and evaporation tendencies. By construction, all three flux correction terms (evaporation, precipitation and circulation) sum up to the total flux correction term.

### 3 Hydrological Cycle Model Development

The development of the new hydrological cycle model of the GREB model is based on the existing zero-order hydrological cycle model of the GREB model. We shortly outline the development of each of the three models and discuss how the change in the reference climatologies from NCEP to ERA-interim has affected the model. All variables are summarised in **Table** 2.



### 3.1 Precipitation

The original GREB precipitation model captures some large-scale aspects of the mean and seasonal cycle of observed precipitation, such as more precipitation in the tropics and warm season over land (Figure 2 and 3). It, however, has substantial differences from the observed precipitation, as it cannot capture the high rainfall in the ITCZ, the enhanced
precipitation over the midlatitudes storm track regions and misses many aspects of the seasonal cycle. The root mean square error for the annual mean of the original GREB model precipitation parameterisation is 1.46 mm/day.

The new parameterisation of precipitation in the GREB model is assumed to be proportional to $q_{air}$, as in the original GREB model. We further assume that relative humidity, $rq$, and upward air motion, $\omega$, increase rainfall. The latter is assumed to be a function of the mean and the standard deviation of the daily mean variation, $\omega_{mean}$ and $\omega_{std}$, respectively. The new
precipitation parameterization is:

$$\Delta q_{precip} = r_{precip} \cdot q_{air} \cdot \left( c_{rq} \cdot rq + c_{\omega} \cdot \omega_{mean} + c_{\omega std} \cdot \omega_{std} \right) \qquad (11)$$

The model parameters, $r_{precip}$, $c_{rq}$, $c_{\omega}$ and $c_{\omega std}$ are fitted to minimise the RMSE between observations and GREB simulated precipitation. The resulting mean precipitation and its seasonal cycle is shown in Figure 2g and 3g. The model is evaluated in a Taylor diagram in Figure 4a and b against observations. We further test the different elements of the
precipitation model by only considering a subset of the variables in Eq. (11), setting the other terms to zero and fitting the parameterizations for these reduced models. This allows us to estimate the effect of each term in the equation, see Figure 4a and b and Figure 5.

Relative humidity ($rq$) is widely used in climate models as a predictor for precipitation (Petoukhov et al., 2005; Petoukhov et al., 1999; Wang & Myask, 2000; Weaver et al., 2001). In the GREB model it increases precipitation mainly over humid
regions like the Amazons basin (Figure 5c and amplifies the seasonal cycle (Figure 5d). The overall pattern of rainfall with high precipitation in the tropics and decreasing towards higher latitudes is not changed. Including $rq$ gives some moderate improvement relative to the original GREB model (Figure 4a comparing marker '0' to marker 'b').

The mean vertical air motion ($\omega_{mean}$) provides a substantial improvement of the precipitation model (Figure 4a and d comparing marker '0' to 'c'). Ascending air masses in the ITCZ lead to increased precipitation, whereas descending air
masses (i.e. in the subtropics) supress precipitation. It creates a sharper and more realistic gradient in precipitation than the original GREB model (compare Figure 3d & 5e). With adding $\omega_{mean}$, GREB is in the range of uncertainty of more complex CMIP5 models in the annual mean and the seasonal cycle (Figure 4a and d).

The GREB precipitation model without $\omega_{std}$ has still fairly weak mean precipitation in the midlatitudes storm track regions (compare Figure 5g and 2g) and has a weak seasonal cycle with the wrong sign in these regions as well (compare Figure 5h
and 3g). The transient pressure systems in these regions lead to large vertical motions ($\omega$) on shorter, daily time scales that result into large precipitation, but have a near zero $\omega_{mean}$. Thus, to capture the precipitation in regions with strong variability in $\omega$, but weak $\omega_{mean}$, we include $\omega_{std}$. This mainly enhances rainfall in the mid- and high latitudes (Figure 2g and 3g).





In summary, the new GREB precipitation model is significantly better than the original model. The RMSE is reduced from by 0.65 mm/day to 0.81 mm /day in the annual mean and by 1 mm/day in the seasonal cycle. GREB precipitation now has a comparable skill to more complex CGCMs and lies within the range of uncertainty CMIP5 modelled precipitation. Introducing the new precipitation parameterisation globally reduces the flux corrections of specific humidity caused by

5 precipitation, see Figure 6c and 6d. The root-mean-square of the flux corrections caused by precipitation are reduced by more than 40%, indicating that the new parametrization has indeed improved the simulation of the hydrological cycle in the GREB model. Similar improvements are gained for the seasonal cycle (Figure 7c and 7d). The original GREB model showed large flux corrections, especially in the tropics where the ITCZ is moving with seasons and in the midlatitudes. The pattern of the flux corrections of the new model still looks similar to the original model, but is only half as large in amplitude

(Figure 6c & d and 7c & d).

### 3.2 Evaporation

In the original GREB model evaporation is calculated using a widely used bulk formula approach (see Eq. (1) in Richter and Xie (2008)). This model does capture the main aspects of the regional differences in the annual mean evaporation in GREB, with enhanced evaporation over subtropical oceans and weaker evaporation over land (Figure 2e). The seasonal cycle

(Figure 3e) is, however, very different from observed and the land-sea differences are too strong.

For the new evaporation model, we retained the original bulk formula approach and included a few minor changes by considering land-sea differences, revised wind ($u_*$) estimates, scaled effectivity and skin temperature. The new evaporation model is:

$$\Delta q_{eva} = \frac{\rho_{air} \cdot c_{eva} \cdot c_w \cdot |u_* + c_{turb}| \cdot v_{soil} \cdot (q_{air} - q_{sat-skin})}{r_{qviwv}} \qquad (12)$$

The constant $c_{eva}$ modifies the evaporation efficiency for a given mean wind speed, $u_*$. $q_{sat-skin}$ is an estimate of saturated humidity considering skin-temperature. It is calculated using:

$$q_{sat-skin} = e^{\frac{z_{topo}}{z_{atmos}}} \cdot 3.75 \cdot 10^{-3} \cdot e^{17.08085 \frac{T_{surf} + c_{eva-temp} - 273.15}{T_{surf} + c_{eva-temp} - 38.975}} \qquad (13)$$

The parameter $c_{eva-temp}$ is a constant temperature offset to mimic skin temperature difference to $T_{surf}$. The parameters $c_{eva}$, $c_{eva-temp}$ and $c_{turb}$ are fitted against observations for ocean and land points individually to minimise the RMSE. The

25 values we estimated are:

$$c_{eva} = \begin{cases} 0.25 \; over \; land \\ 0.58 \; over \; ocean \end{cases} \qquad (14)$$

$$c_{eva-temp} = \begin{cases} 5 \; K \; over \; land \\ 1 \; K \; over \; ocean \end{cases} \qquad (15)$$

$$c_{turb} = \begin{cases} 11.5 \; over \; land \\ 5.4 \; over \; ocean \end{cases} \qquad (16)$$





The scaled effectivity ($c_{eva}$) is lower over land than over oceans reflecting the fact that for a given $u_*$ more evaporation is simulated over oceans. This appears to be realistic considering that land has lower wind speeds near the surface for a given $u_*$ due to the topography and vegetation. The value of $c_{eva} \cdot c_w$ closely match the observed values over oceans (Anderson & Smith, 1981; Merlivat, 1978).

The skin temperature difference approximated by $c_{eva-temp}$ is larger over land. It reflects that the GREB model does not simulate the daily cycle and the larger daily cycle over land leads to an effectively larger difference between the simulated $T_{surf}$ and the skin temperature. The offset of 1$^{o}$C over oceans is also found by (Feng et al., 2018).

The wind magnitudes ($u_*$) in the original GREB was estimated on the basis of the monthly mean climatologies of the zonal and meridional wind components. This, however, is not an accurate estimate of the monthly mean wind magnitudes, as it

neglects the turbulent term due to high frequent variability. In the new GREB model we estimate the monthly mean $u_*$ climatology based on the original 6 hourly ERA-Interim time steps.

We can estimate how much each of the changes has improved the evaporation model by including only one of these changes and fitting the parameters of these models individually, see Figure 4 b & e and Figure 8.

Fitting the evaporation efficiency $c_{eva}$ and the turbulent wind factor improves evaporation over land, especially in the

seasonal cycle (Figure 8d) and reduces the strength of evaporation over the ocean. The increase in evaporation over land is caused by the increase in the turbulent wind factor. $c_{eva}$ would decrease the evaporation in the annual mean and the seasonal cycle. By including the new estimate of monthly mean wind speed $u_*$ the pattern of evaporation is getting closer to observations, especially over the oceans (i.e. Figure 8f, North Atlantic) and by including the new estimate of skin temperature the seasonal cycle is improving slightly (Figure 4e).

The original GREB model was evaporating too much on the annual mean (see Figure 2e) especially over the equatorial pacific and Atlantic. The new hydrological cycle parameterisation largely decreases evaporation over these regions and the flux corrections are reduced over the globe in the annual mean (Figure 6e & f). The correlation of the annual mean experiences the largest changes from changing the reference climatology (Figure 4b).

In the seasonal cycle, each included variable improves the simulation of evaporation in GREB (Figure 4e). The seasonal

cycle of flux corrections caused by evaporation in the original GREB model are large over land and large over oceans. There are positive flux corrections around the equator and negative flux corrections over the oceans north of the equator (Figure 7e). The improved evaporation seasonal cycle mainly removes this distinct pattern over the oceans and reduces flux corrections over most land areas. (Figure 7e & f). Overall, the new evaporation model is slightly better than in the original GREB model, but it still has substantial limitation in simulating the seasonal cycle correctly (Figure 2h & 3h).





### 3.3 Transport

The original GREB model transport of moisture was very weak and had little agreement with observations (Figs. 2f and 3f). Atmospheric transport of moisture in GREB (Eq. (4)) is controlled by diffusion and advection with mean winds. This model considered a divergence free 2-dimensional flow.

However, moisture convergence, as it occurs for example in the ITCZ, is important for the transport of moisture in these regions. The mean convergence by advection including the moisture convergence term is:

$$\overline{\vec{\nabla}(\vec{u} \cdot q_{air})} = \overline{\vec{u} \cdot \vec{\nabla} q_{air}} + \overline{q_{air} \cdot \vec{\nabla}\vec{u}} \tag{17}$$

The second term on the RHS was not considered in the original GREB model, but is now considered in the new model. The moisture convergence term can be approximated by knowing the vertical air flow assuming continuity and hydrostatic
balance:

$$\overline{q_{air} \cdot \vec{\nabla}\vec{u}} \approx q_{air} \cdot f \cdot \frac{dt_{crcl}}{z_{vapour} \cdot \rho_{air} \cdot g} \cdot (-\omega) \tag{18}$$

with the known parameters scaling height of water vapour, $z_{vapour}$, density of air, $\rho_{air}$, gravitational acceleration, $g$, and the circulation time step, $dt_{crcl}$. The scaling factor, $f$, should theoretically be 1.0, but the mean large-scale horizontal winds and vertical velocities may not perfectly match. A fit of Eq. (18) to observations finds the $f = 2.5$.

This new model has now a fairly realistic transport in the annual mean and the seasonal cycle (Figure 2i and 3i), with clear moisture transport out of regions with diverging flow (e.g. in the subtropics of coast of Peru) and into converging zones (e.g. ITCZ). The new parameterisation of convergence also reduces the flux corrections in the annual mean and the seasonal cycle (Figs. 6 g & h and 7 g & h).

### 3.4 Boundary Conditions and Input Data

The original GREB model used the NCEP reanalysis as boundary conditions and as references for estimating the parameterization of the model. New generations of reanalysis products have improved, because of the use of better models, better input data and better assimilation products (Dee et al., 2011). This is shown by Chen and Liu (2016) who investigated the variability and trends of the vertically integrated water vapour and found that ECMWF's ERA-Interim reanalysis has a higher accuracy than NCEP and a better agreement with observations over oceans and in the tropics. NCEP underestimates
water vapour in troposphere (Kishore et al., 2011) and Mooney et al. (2011) found a higher correlation of surface temperature in ERA-Interim to observations then NCEP in Ireland. We therefore changed the reference climatology of specific humidity in GREB from NCEP to ERA-Interim. To get a consistent model we also take surface temperature, horizontal winds the climatology of omega and standard deviation of omega from ERA-Interim. The parameters of our new GREB hydrological cycle model are then fitted against the new reference climatologies.



We estimate the effect that the change in reference climatologies have on the new GREB hydrological cycle by fitting the parameters of the new model as described above to both the NCEP and ERA-interim reanalysis. The resulting hydrological cycle models are evaluated against observations (GCPC and ERA-interim) in Taylor diagrams for the annual mean

This doesn't lead to major improvements in the representation of the hydrological cycle in the GREB model however, it

increases the correlation of precipitation, evaporation and circulation and reduces the RMSE (Figure S1 in the supplementary plots). The main improvement is in the tropics and might be related to the underestimated value of specific humidity in the tropics found by (Chen & Liu, 2016; Kishore et al., 2011).

## 4 Model Verification

We now test the new hydrological model in a series of three different sensitivity experiments. The focus in the discussion

will be on evaluating the new model. We will leave more in-depth analysis of some of these experiments to future studies.

### 4.1 Seasonal Cycle

The response of the hydrological cycle to seasonal changes is a good test for evaluating the skill of the hydrological cycle model. The GREB model applies monthly flux correction terms to maintain a mean atmospheric humidity as observed. Thus, by construction the specific humidity in each calendar month in the GREB model will be identical to the observations, see

Figure 9a.

To illustrate that the seasonal cycle is not a feature of the seasonally varying flux corrections we changed the flux corrections to an annual mean value for the original GREB model (middle column in Figure 9) and for the new GREB model (right column in Figure 9). This annual mean flux correction value is added on every time step to the tendency equation of specific humidity (Eq. (4)).

With the new parameterisations for precipitation, evaporation and circulation the new GREB model resolves the seasonal cycle better than the original GREB model (Figure 9). The seasonal cycle of the original GREB model was too weak in the northern hemisphere when compared to observations and throughout the year GREB was too dry (Figure 9b). For the southern hemisphere, the original GREB was too wet. The new GREB model captures the high humidity in northern hemispheric summer and the low values in winter (Figure 9c). This makes the seasonal cycle stronger in the new GREB and

it is closer to the reference climatology. In summary, the new GREB hydrological cycle model simulates the seasonal evolution of the atmospheric humidity very well and significantly better than the original GREB model.

### 4.2 El Nino Southern Oscillation

Strong El Nino and La Nina events lead to significant changes in the tropical precipitation and associated hydrological cycle changes. Since these natural modes of climate variability are well documented they present a good test case for the GREB

model.



We therefore conducted a set of sensitivity experiments with the GREB model forced by the mean conditions for strong El Nino and La Nina events. The GREB model was forced with mean composites of $T_{surf}$, horizontal winds and omega from observations for four El Nino (1982/83, 87/88, 91/92, 97/98) and La Nina (1988/89, 99/00, 07/08, 10/11) events. The anomalies are calculated around El Nino/La Nina from May before the peak in December to April in the following year and

5 against the climatological mean. In the GREB model simulation they are added on top of the reference climatology. The observed anomalies in the hydrological cycle during these ENSO events are shown in Figure 10a-c. We clearly note strong regional changes in the precipitation in the tropical Pacific that match changes in moisture transport (Figure 10c), illustrating that ENSO events mark strong regional changes in the hydrological cycle related to changes in the circulation.

The new GREB response in precipitation shows a strong similarity with the observed changes (Figure 10g). There is a shift

of rainfall from the Maritime Continent towards the NINO3.4 region (5°N to 5°S & 170°W to 120°W) over the Pacific. However, the overall amplitude in the precipitation response is somewhat weaker than observed. In contrast, the original GREB model has nearly no precipitation response to the ENSO forcings. This is consistent with the weak response in the circulation in the original GREB model (Figure 10f). The correlation between the GREB simulated El Nino response increases from 0.0 for the original GREB model to 0.9 with the new GREB model.

The observed evaporation response to ENSO events in the tropical Pacific is somewhat counteracting the precipitation response, as we observe mostly decreased evaporation over regions with enhanced precipitation and increased evaporation over regions with reduced precipitation (Figure 10a and b). These evaporation changes are mostly caused by changes in winds, with decreased evaporation over regions were the winds have weakened (e.g. NINO3.4 region). The new GREB model somewhat captures this pattern, but shows a stronger evaporation response, which partly explains the weaker

precipitation response. However, both the original and the new GREB model evaporation have only a weak spatial correlation (0.3) with the observed evaporation changes overall.

The observed strong changes in the circulation of atmospheric humidity (Figure 10c) is mostly due to changes in the convergence of moisture (e.g. omega). Since, convergence of moisture was not considered in the original GREB model, the simulated changes in the circulation are very weak in the original GREB model (Figure 10f). The new GREB model does

consider convergence of moisture and simulates the changes in the circulation of atmospheric humidity very similar to the observed (Figure 10i). The new circulation parameterisation in GREB improves the correlation between the observed and the simulated circulation tendency from 0.3 (original GREB) to 0.95.

In summary, new GREB model does simulate the precipitation and circulation response to ENSO conditions fairly well, whereas the original GREB model had very little skill, illustrating the significant improvement of the new GREB model over

30 the original GREB model. However, the evaporation response in both models is not as well simulated as the precipitation and circulation response.



### 4.3 Global Warming

The response of the hydrological cycle to global warming is one of the potential applications of the GREB model and a comparison of the GREB model with the CMIP model simulations response to global warming provides a good test. The CMIP5 ensemble mean response of precipitation shows a distinct increase of rainfall in the equatorial pacific, decreases of

mean rainfall in some subtropical regions (i.e. east pacific) and increases in some areas of the midlatitudes, see Figure 11a. This pattern is normally referred to as wet-get-wetter paradigm (Held & Soden, 2006). Although this approach has been questioned by more recent studies (Chadwick et al., 2013) it still gives a good first order approach of the changes in the global hydrological cycle, although changes over land might be muted or even reversed (He & Soden, 2016).

To evaluate the GREB hydrological cycle model independent of the other GREB model components, such as the $T_{surf}$

tendencies, we force the original and new GREB model with RCP8.5 equivalent $CO_2$ concentrations and all other input variables for the hydrological cycle model taken from CMIP model simulations. That is, we prescribe $T_{surf}$, horizontal winds and vertical velocity in the scenario run taken from the RCP8.5 CMIP5 ensemble mean of the models described in **Table 1**. In the control run the reference boundary conditions of $T_{surf}$, horizontal winds and omega are taken.

The precipitation response in the original GREB model is positive in all locations and it closely follows the pattern of

specific humidity in the control simulation (see Eq. (1) and Figure 11d). This is mainly due to an increase in the saturation water vapour pressure of about 7% per degree of warming (Clausius-Calpeyron). The original GREB precipitation response pattern is not correlated to the CMIP5 ensemble mean response pattern (Figure 12a), suggesting that local differences in the precipitation response are very different from those in the CMIP simulations.

The improved GREB model response pattern is similar to the CMIP models with enhanced and reduced response roughly at

similar locations. This is strongly related to the moisture transport changes.  However, the overall global mean precipitation response in the new GREB model is shifted upwards compared to the CMIP5 ensemble mean, which is related to the much stronger response in evaporation. In CMIP5 models, we see a muted response of evaporation mainly due to changes in surface relative humidity and surface stability (Richter & Xie, 2008).

### Summary and Discussion

In this study, we introduced the newly develop hydrological cycle model for the GREB model. It consists of three parts: precipitation, evaporation and transport. The development of these models started from the existing zero order hydrological cycle of the GREB model and used physical reasoning and observations for fitting parameters.

The simulation of precipitation and transport of moisture in the new hydrological cycle model is now comparable in skill to CMIP model in terms of annual mean and the seasonal cycle of rainfall. The evaporation has only improved slightly, but

does simulate the annual mean values fairly well. However, it is still different from the observed seasonal cycle and the skill is much lower than that of the CMIP model. This suggests that the evaporation model is still a limiting factor in the GREB model.



We applied the new hydrological cycle model to a number of sensitivity studies, that illustrated that the new hydrological cycle model is much improved over the original GREB model. The annual cycle simulation without any correction terms is very realistic with the new model, the precipitation response to ENSO events is now very similar to the observed, owing to the much-improved transport of moisture. Finally, the response to global warming now shows a precipitation response

pattern that is comparable that of CMIP models. Again, a limiting factor in this sensitivity experiment was the evaporation response of the GREB model in comparison to that of CMIP models.

An interesting aspect of the GREB model is that it has the atmospheric circulation (vertical and horizontal winds), humidity and surface temperatures as boundary conditions. This allows the GREB model to be used as a diagnostic tool to understand how different boundary conditions affect aspects of the climate system, such as the hydrological cycle's response to global

warming. It may also help to study how biases in the hydrological cycle in CMIP models related to different boundary conditions from the atmosphere, such as biases in the vertical winds. The new GREB hydrological cycle is therefore a good tool in helping to conceptually understand the hydrological cycle and its response to global warming or other external forcings. It will further help in understanding CMIP model biases in the simulation of the hydrological cycle.

**Code availability**

The GREB model source code used in this paper as well as the data used to run the model is available on GitHub: https://github.com/christianstassen/greb-hydro-develop-gmd.git. The GitHub repository contains detailed documentation on how to download the source code and installation instructions along with an example script on how to plot data obtained from GREB model simulations. The GREB source code is tested on recent-generation Mac platforms.

**Acknowledgments**

This study was supported by the Australian Research Council (ARC), with additional support coming via the ARC Centre of Excellence in Climate System Science and the ARC Centre of Excellence in Climate Extremes.

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





**Tables**

**Table 1: List of CMIP 5 models.**

| Models | | |
|---|---|---|
| ACCESS1-0 | CSIRO-Mk3-6-0 | HadGEM2-ES |
| ACCESS1-3 | CanESM2 | inmcm4 |
| bcc-csm1-1 | EC-EARTH | MIROC-ESM-CHEM |
| bcc-csm1-1-m | FGOALS-g2 | MIROC-ESM |
| BNU-ESM | FGOALS-s2 | MIROC4h |
| CCSM4 | FIO-ESM | MIROC5 |
| CESM1-BGC | GFDL-CM3 | MPI-ESM-LR |
| CESM1-CAM5 | GFDL-ESM2G | MPI-ESM-MR |
| CESM1-FASTCHEM | GFDL-ESM2M | MPI-ESM-P |
| CESM1-WACCM | GISS-E2-H-CC | MRI-CGCM3 |
| CMCC-CM | GISS-E2-H-R | NorESM1-M |
| CMCC-CM5 | HadGEM2-CC | NorESM1-ME |

**Table 2: Variables of the GREB model**

| Variable | Dimension | Description |
|---|---|---|
| $c_{eva}$ | constant | Evaporation efficiency |
| $c_{eva-temp}$ | constant | Temperature scaling of evaporation |
| $c_{turb}$ | constant | Turbulent wind offset for evaporation |
| $c_{rq}$ | constant | Precipitation parameter for rel. humidity |
| $c_{\omega}$ | constant | Precipitation parameter for omega |
| $c_{\omega std}$ | constant | Precipitation parameter for standard deviation of omega |
| $f$ | constant | convergence scaling parameter |
| $g$ | constant | gravitational acceleration |
| $q_{air}$ | x, y, t | atmospheric humidity |
| $q_{sat}$ | x, y, t | saturation pressure |
| $q_{sat-skin}$ | x, y, t | saturation pressure with temperature offset |
| $r_{precip}$ | constant | mean lifetime of water vapour |
| $r_{qviwv}$ | constant | regression between atm. humidity and vertically |



| | | integrated water vapour |
|---|---|---|
| $rq$ | x, y, t | relative humidity |
| $T_{surf}$ | x, y, t | surface temperature |
| $|\vec{u}_*|$ | x, y, t | Absolute wind climatology |
| $\vec{u}$ | x, y, t | Horizontal wind climatology |
| $z_{atmos}$ | constant | Scaling height of atmosphere |
| $z_{topo}$ | x, y, t | Topographic height |
| $z_{vapour}$ | constant | Scaling height of water vapour |
| $\vartheta_{soil}$ | x, y, t | Surface wetness fraction |
| $\rho_{air}$ | constant | Density of air |
| $\omega_{std}$ | x, y, t | Standard deviation of vertical wind climatology |
| $\Delta q_{eva}$ | x, y, t | mass flux for the atmospheric humidity by evaporation |
| $\Delta q_{precip}$ | x, y, t | mass flux for the atmospheric humidity by precipitation |
| $\Delta q_{correct}$ | x, y, t | Mass flux correction of specific humidity |
| $\Delta q_{cor-circul}$ | x, y, t | Mass flux correction due to circulation |
| $\Delta q_{cor-evapo}$ | x, y, t | Mass flux correction due to evaporation |
| $\Delta q_{cor-precip}$ | x, y, t | Mass flux correction due to precipitation |
| $\Delta q_{precip-GREB}$ | x, y, t | Precipitation change in GREB |
| $\Delta q_{precip-OBS}$ | x, y, t | Precipitation change in observations |
| $\Delta t$ | constant | Model integration time step |
| $dt_{crcl}$ | constant | Model integration time step for circulation |
| $\kappa$ | constant | Isotropic diffusion coefficient |
| $\omega$ | x, y, t | Vertical velocity in pressure coordinates |





**Figures**

Figure 1: **GREB mean state boundary conditions and reference climatologies: topography (a), surface temperature (b), surface humidity (c), 850 hPa winds (d), vertical velocity omega (e) and the daily standard deviation of vertical velocity omega (f).**





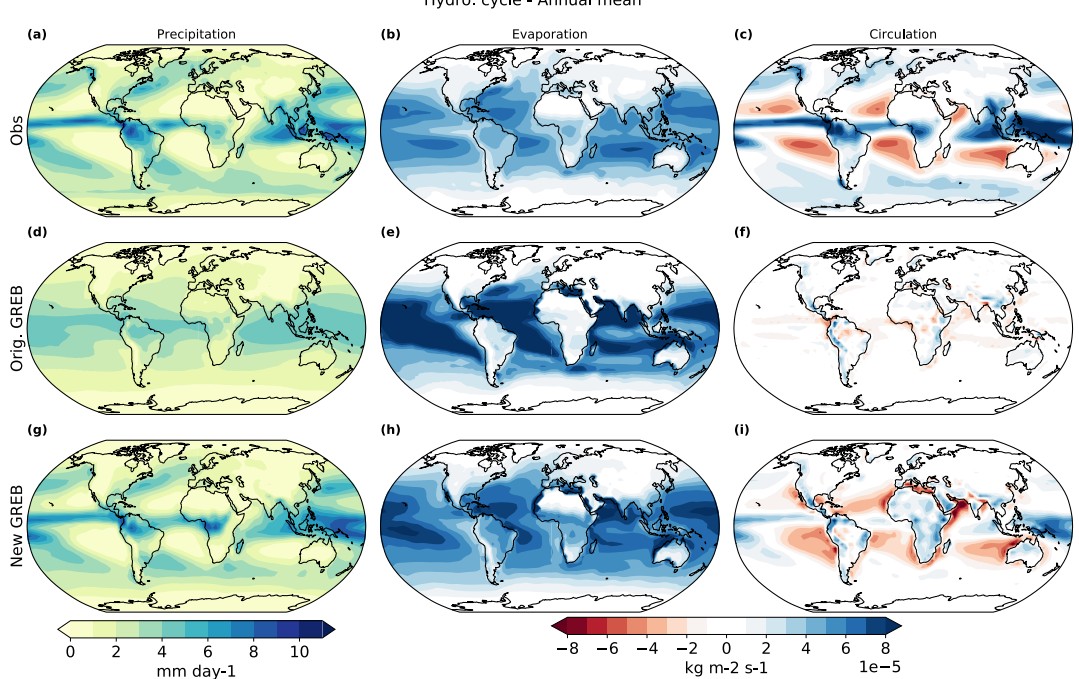

**Figure 2:** The decomposition of the hydrological cycle into its parts precipitation in mm/day (left column), evaporation (middle column) and circulation in kg/m2/s (right column) in observations (upper row), the original GREB model (middle row) and the new GREB model (lower row) for the annual mean.



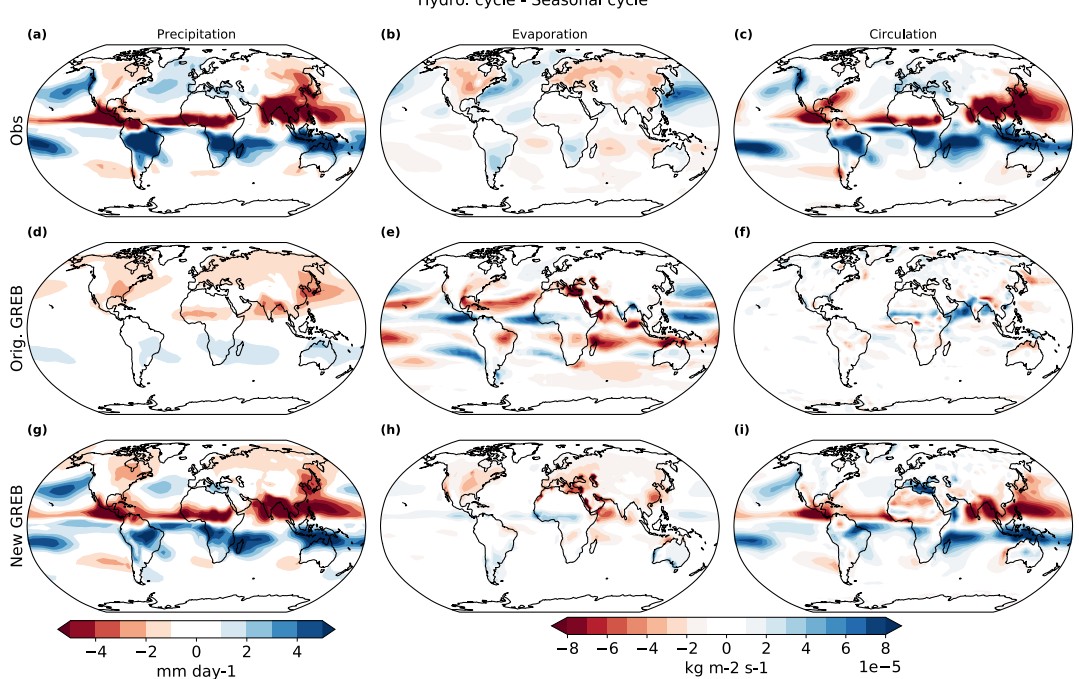

**Figure 3:** **As Fig. 2 but for the seasonal cycle (DJF-JJA). The decomposition of the hydrological cycle into its parts precipitation in mm/day (left column), evaporation (middle column) and circulation in kg/m2/s (right column) in observations (upper row), the original GREB model (middle row) and the new GREB model (lower row) for the seasonal cycle.**





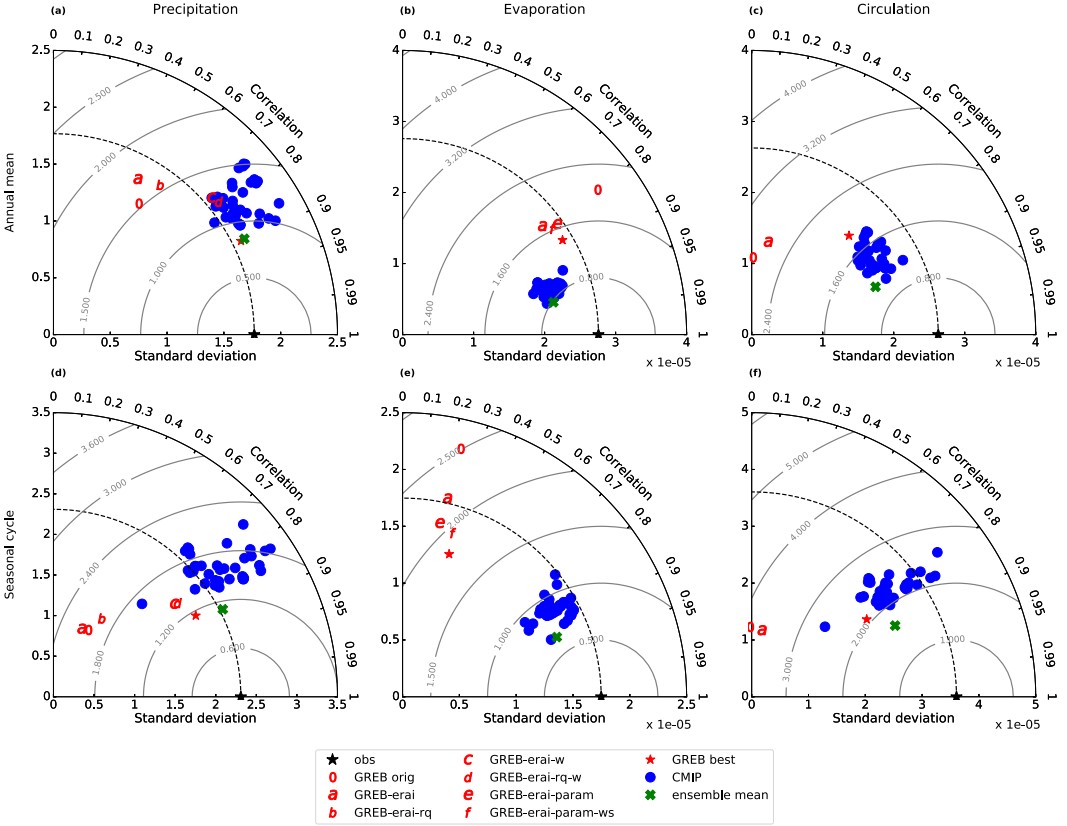

**Figure 4: Precipitation (left column), evaporation (middle column) and circulation (right column) in the annual mean (top row) and seasonal cycle (bottom row) in mm/day in a Taylor diagram against observations from GPCP and ERA-Interim. Red colours indicate different GREB parametrisations with 0 being the original and * the best parametrisation. Blue dots are pi-Control CMIP5 models and the green cross indicates the ensemble mean of all CMIP5 models.**

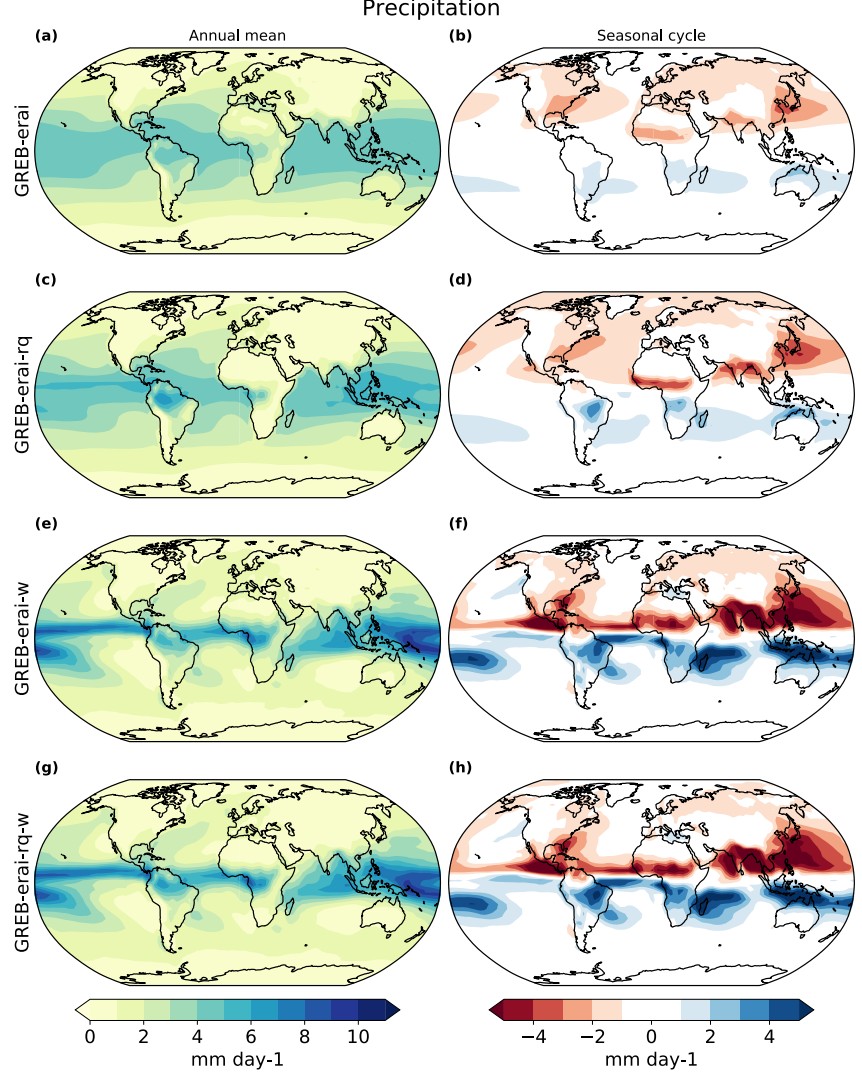

**Figure 5:** **Annual mean precipitation for four development steps of the GREB precipitation parametrisation (a, c, e, g) and their corresponding seasonal cycles (b, d, f, h) in mm/day. The first step was changing the specific humidity boundary climatology (a) and (b). Then subsequently more variables have been added to the precipitation parametrisation: adding only relative humidity (c, d), adding only omega (e, f), adding relative humidity and omega (g, h).**





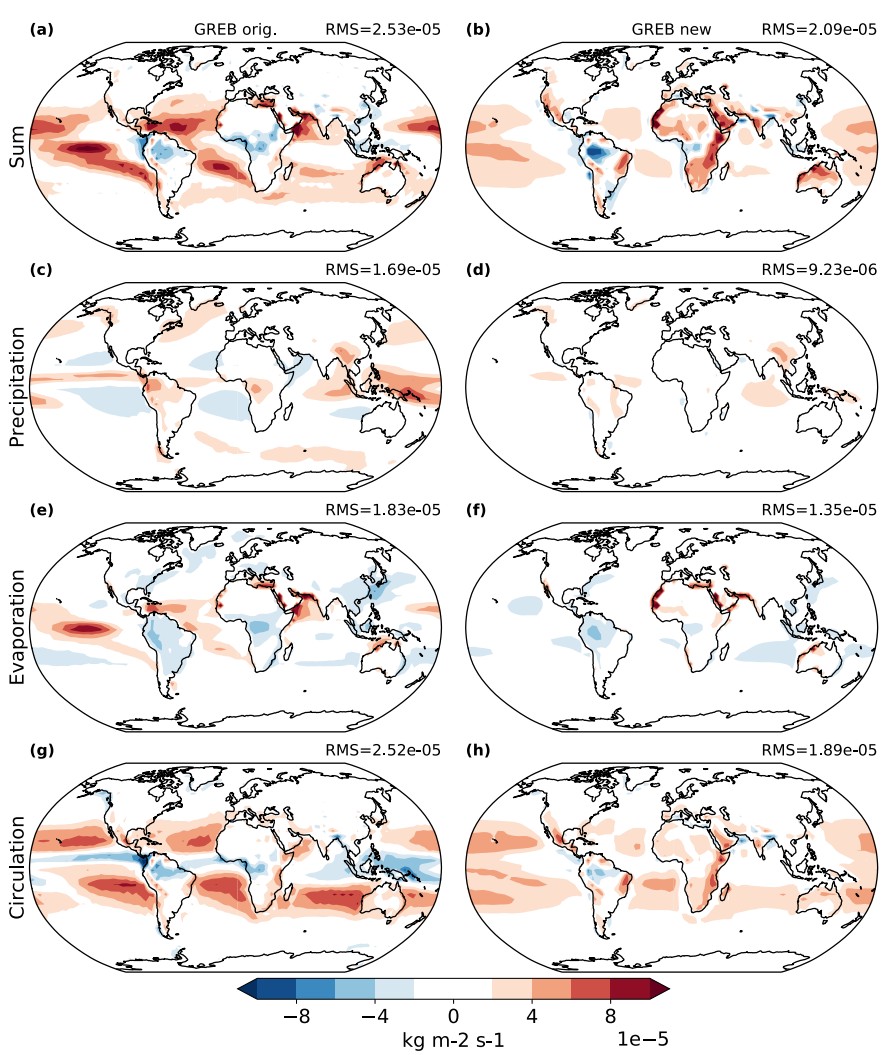

**Figure 6:** Annual mean flux corrections of specific humidity for the original GREB model (a) and the improved GREB model (b). The flux corrections are then split into their contributions of precipitation (c, d), evaporation (e, f) and circulation (g, h) for the original GREB model (left column) and the improved GREB model (right column) in kg/m2/s. The top right shows the global root-mean-square (RMS).



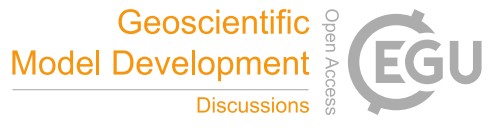

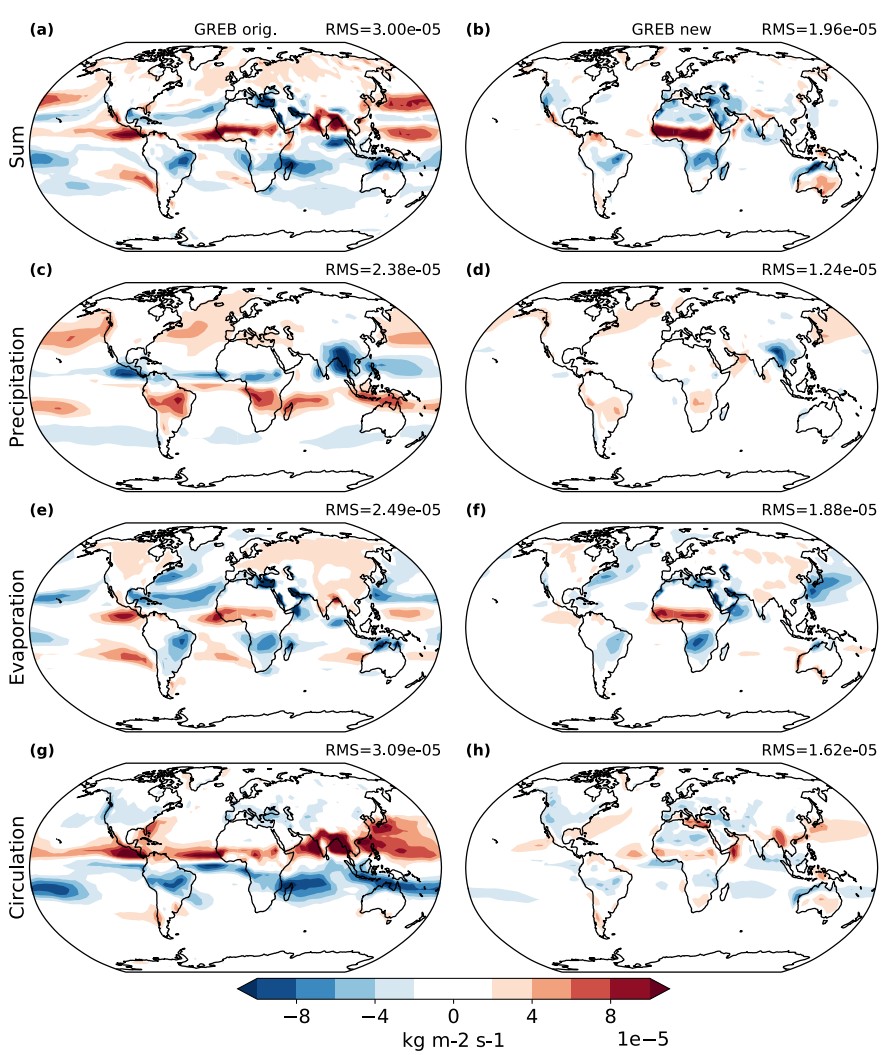

**Figure 7:** **As Fig. 6 but for the seasonal cycle (DJF-JJA). Flux corrections of specific humidity for the original GREB model (a) and the improved GREB model (b). The flux corrections are then split into their contributions of precipitation (c, d), evaporation (e, f) and circulation (g, h) for the original GREB model (left column) and the improved GREB model (right column) in kg/m2/s. The top right shows the global root-mean-square (RMS).**


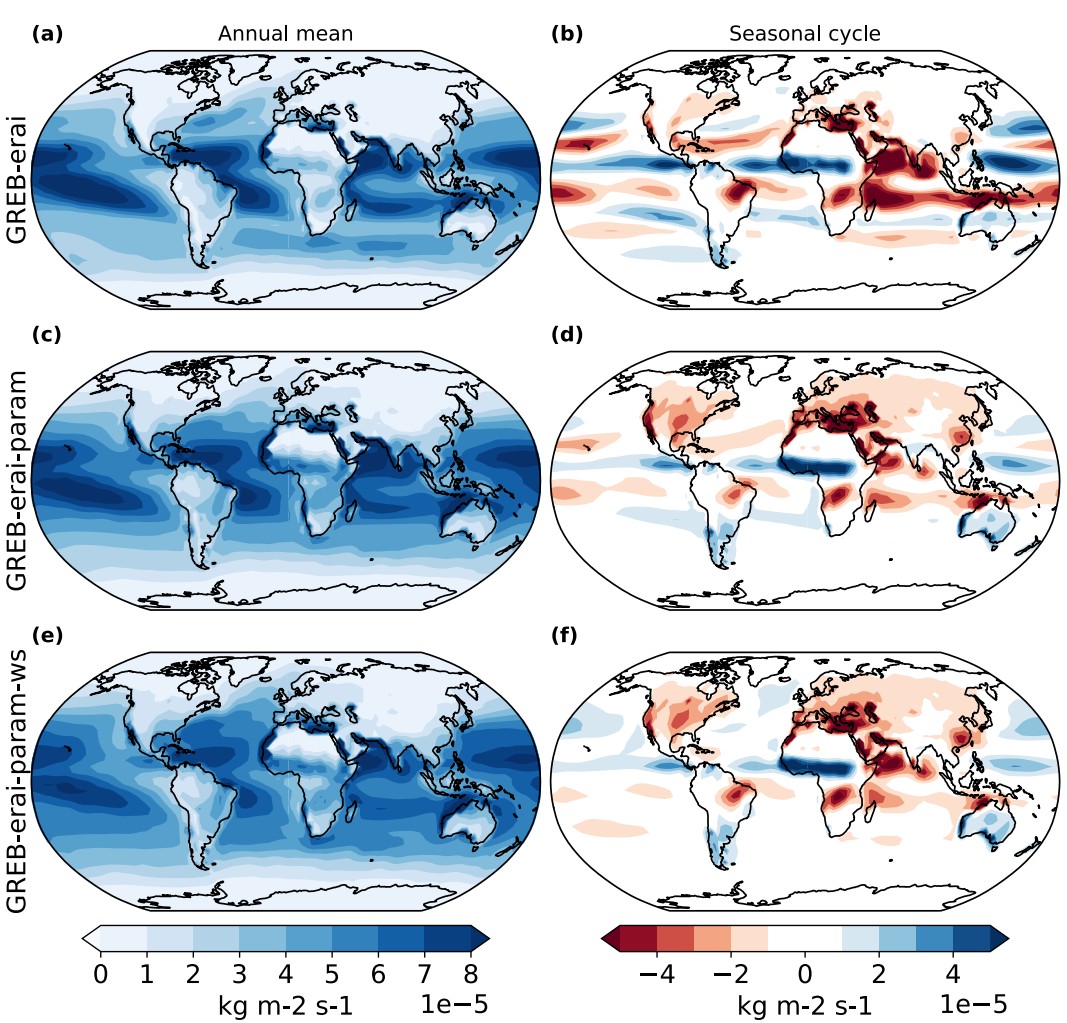

**Figure 8:** Annual mean evaporation for three development steps of the GREB evaporation parametrisation (a, c, e) and their corresponding seasonal cycles (b, d, f) in kg/m2/s. The first step was changing the boundary climatology (a) and (b). Then subsequently more variables have been added to the evaporation parametrisation: fitting the evaporation parameters separately for ocean and land (c, d) and fitting parameters and prescribing the wind speed (e, f).



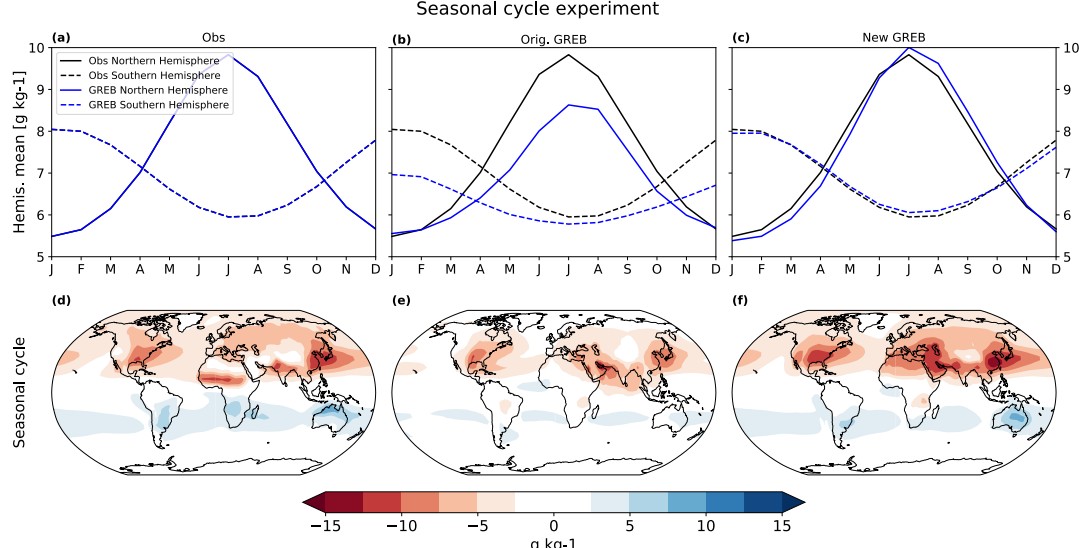

**Figure 9:** **Annual cycle of specific humidity with seasonal varying flux corrections (a, d) and annual mean flux corrections for original GREB (b, e) and improved GREB (c, f) in g/kg. The top row shows the northern (solid) and southern (dashed) hemispheric mean for observations (black) and GREB (blue). The bottom shows the respective seasonal cycle (DJF-JJA). For the seasonally varying flux corrections (a) GREB (blue) matches observations (black).**




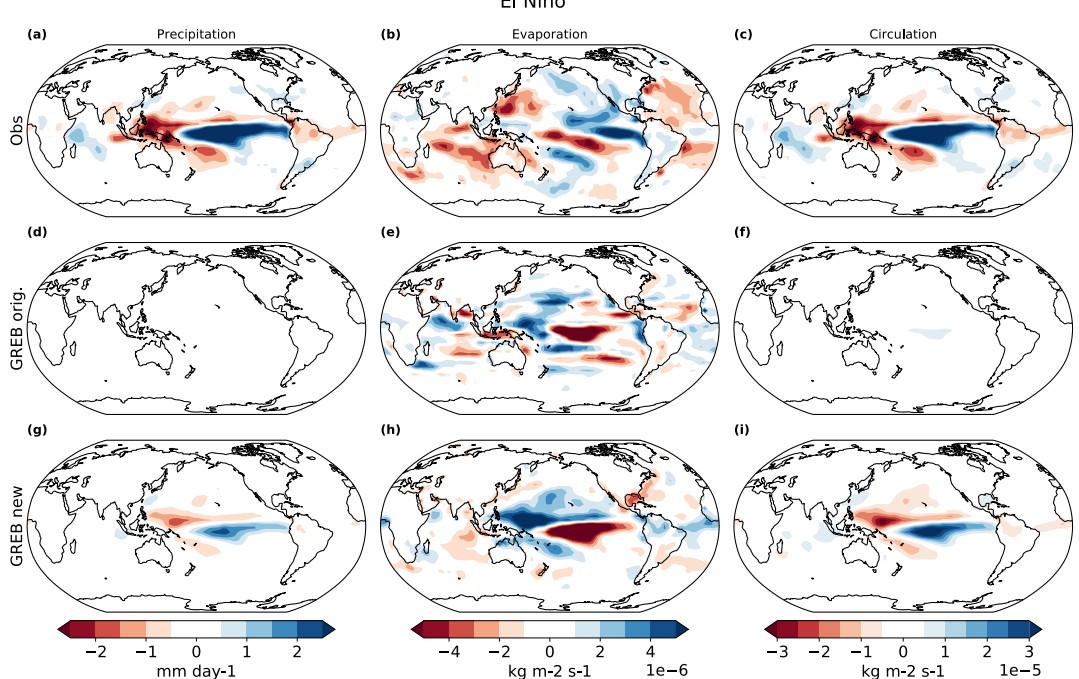

**Figure 10:** **The El Nino response of the hydrological cycle in: observations for precipitation (a) in mm/day, evaporation (b) and circulation (c) in kg/m2/s (upper), original GREB model for precipitation (d), evaporation (e) and circulation (f) (middle) and the improved GREB model for precipitation (g), evaporation (h) and circulation (i) (lower). GREB uses prescribed anomalies from an El Nino composite mean of surface temperature, horizontal winds and vertical winds (omega).**





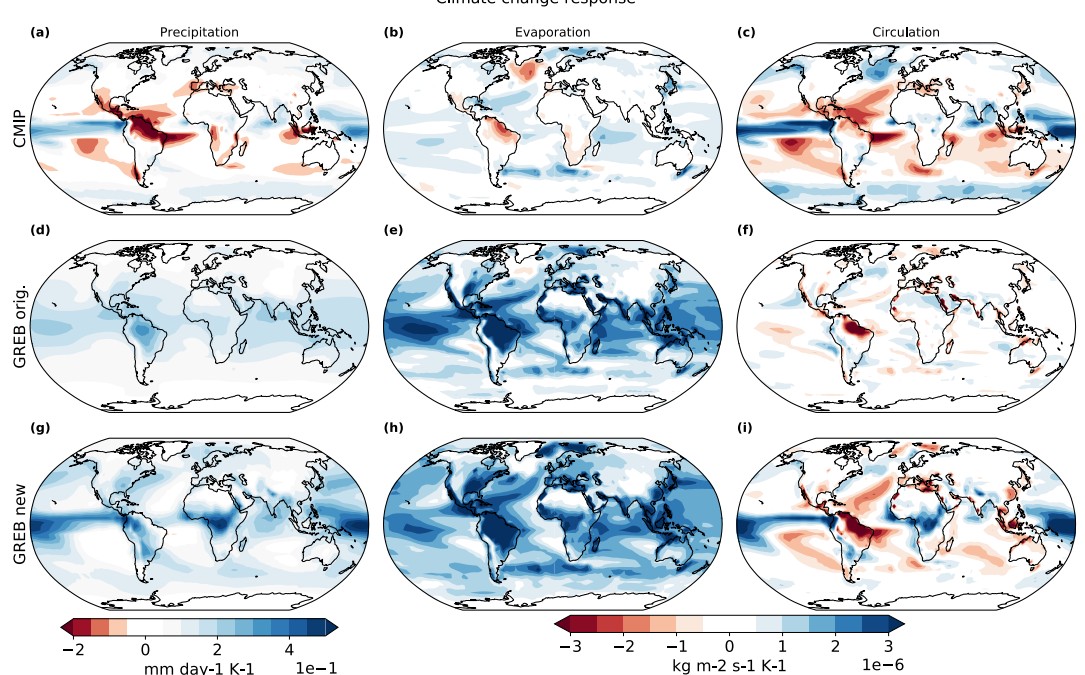

**Figure 11:** **Response of the hydrological cycle to an RCP8.5 forcing in the: CMIP5 ensemble mean for precipitation (a) in mm/day, evaporation (b) and circulation (c) in kg/m2/s (upper), original GREB model for precipitation (d), evaporation (e) and circulation (f) (middle) and the improved GREB model for precipitation (g), evaporation (h) and circulation (i) (lower). GREB uses pre-**
5 **scribed anomalies from CMIP5 ensemble mean of surface temperature, horizontal winds and vertical winds (omega). All responses are shown per degree of warming.**





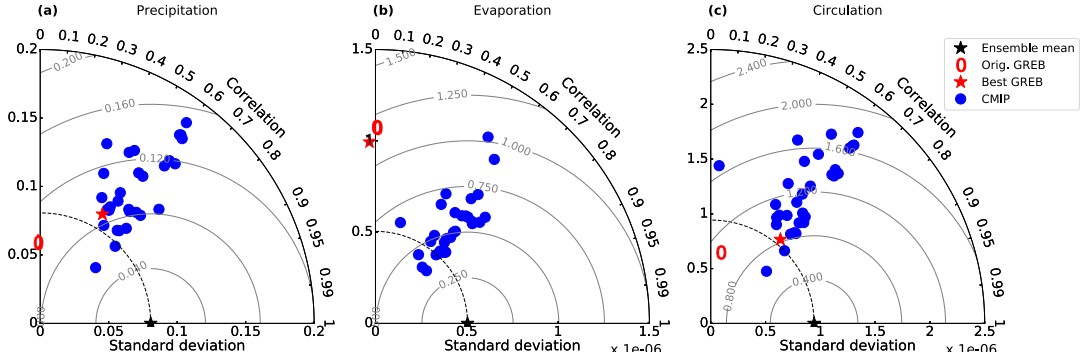

**Figure 12:** **RCP8.5 response of CMIP5 models (blue), original GREB (0) and improved GREB (*) per degree of global warming against the CMIP5 ensemble mean (black star). Precipitation is shown on the left, evaporation in the middle and circulation on the right column. GREB uses prescribed anomalies from the CMIP5 ensemble mean of surface temperature, horizontal winds and vertical winds (omega). The correlation of the original GREB model precipitation response with the ensemble mean is zero. The original and improved GREB model have zero correlation with the ensemble mean evaporation and the standard deviation is one for both.**





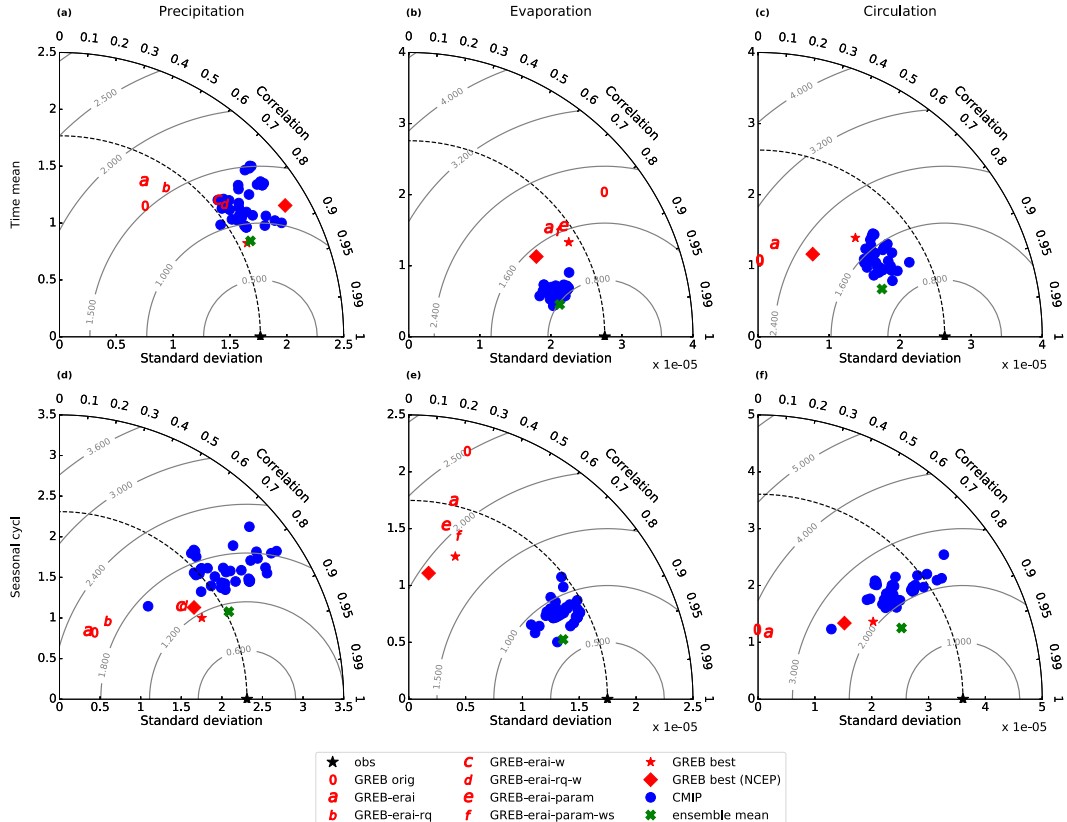

**Figure S1:** **Precipitation (left column), evaporation (middle column) and circulation (right column) in the annual mean (top row) and seasonal cycle (bottom row) in mm/day in a Taylor diagram against observations. Red colours indicate different GREB parametrisations with 0 being the original and \* and diamond the best parametrisation. Star (\*) is the best model for the ERA-Interim boundary conditions and diamond uses the NCEP boundary conditions. Blue dots are CMIP5 models and the green cross indicates the ensemble mean of all CMIP5 models.**