# Peer review of "A Hydrological Cycle Model for the Globally Resolved Energy Balance Model (GREB) v1.0"

_Geoscientific Model Development, 2018_

## Referee Comment (RC1) · Anonymous Referee #1 · 18 Jul 2018

This study contributes to the development of a new hydrological cycle model that is found able to improve precipitation, evaporation, and water vapor transport in idealized simulations. The development framework is reasonable as it incorporates additional parameters with physical mechanisms explained. A series of analyses are performed to illustrate the improvements, and model verifications shed lights on further applications. Therefore, I support to publish this work on *Geoscientific Model Development*. Comments and suggestions are summarized below to potentially improve the manuscript.

**General Comments**

Can the authors elaborate the difference between the GREB model and CMIP5 CGCMs? As atmospheric and oceanic circulations are not simulated in the GREB model, it is probable that the GREB simulations give rise to results lacking of dynamical contribution. Does this lacking component play some roles in affecting the performance in the new model?

Due to the fact that CMIP5 CGCMs have biases in simulating circulations (e.g., Yang et al., 2018, *Journal of Climate*), the differences, at least for circulations, between the GREB model and CMIP5 models could be traced to the differences between prescribed wind fields in ERA-Interim reanalysis product and simulated wind fields. In other words, different background wind states may be part of the reason generating the discrepancy. What if comparing results using the CMIP5 simulated mean fields to force the old/new GREB model? Do authors have insights toward this point?

If no daily weather systems are simulated, does that mean the temporal integration is performed in the time step of one month or more in the GREB model? I am confused because the model time step is 12hrs as mentioned in Page 3 Line 14. In addition, having no weather system simulated does not mean no internal variability generated in the model. I suggest authors rephrasing this pragraph or providing further explanation.

**Specific Comments**

Page 1 Line 8: Does "the hydrological cycle" refer to the hydrological model? Similar mixture appears throughout the abstract.

Page 1 Line 9: The authors should clarify the meaning of "zero order". Does the "order" means numerical convergence rate in time or space?

Page 2 Line 6: Authors may consider mentioning the computational efficiency of idealized model here.

Page 2 Line 23: I suggest re-organizing the paragraphs that describe the GREB model. For example, the model layer configuration and resolution in Page 3 Line 10 can be introduced before the description of the NCEP climatological fields used in the original GREB model. This may make introduction of the GREB model framework smoother.

Page 2 Lines 25-26: Any specific reason generating topography from an atmospheric model? Why not using ETOPO dataset?

Page 2 Line 29: I suggest providing brief explanation for the reason of changing dataset here and directing further details to section 3.4. Also, the NCEP reanalysis datasets are used during 1950-2008, whereas the ERA-Interim reanalysis data during 1979-2015. Therefore, long-term mean climatology values may be different. What is the results using the NCEP data during 1979-2015?

Page 4 Line 3: How important is the diffusion term compared to advection? Could we ignore diffusion effect in large-scale circulation dynamics and/or thermodynamics?

Page 10 Line 6: ENSO events include both El Niño and La Niña. Is the analysis for La Niña shown in the manuscript? Figure 10 only shows analysis for El Niño composite, but not La Niña. How does response to La Niña look like?

Page 10 Line 9: Figure 10g shows the improvement in precipitation anomalies. It

could be informative also discussing extratropical precipitation response because ENSO-mid-latitude linkages were also well documented. In Figure 10g, no precipitation increases in the Southern U.S. region. Does that indicate ENSO teleconnection is still not resolved well in the new model?

**Technical Corrections**

Page 1 Line 15: It is better to mention the full name of "CMIP" in the abstract.

Page 1 Line 16: El Nino → El Niño (and through the manuscript)

Page 1 Line 17: Add the full name of "CGCM".

Page 1 Line 25: What is "CGCMs" stands for? If it refers to "General Circulation Models" at the end of Line 24, it should be abbreviated as GCMs.

Page 1 Line 26: "(AR4)" only appears once here, there's no need to provide abbreviation.

Page 3 Line 5: Figure 2c and 3c → Figures 2c and 3c. I found this kind of error appears through the manuscript (e.g., Page 4 Line 4 and Page 5 Line 23). Please check carefully and revise them consistently.

Page 3 Line 7: rcp85 → RCP8.5 (to be consistent to that in the caption of Figure 11, Page 27).

Page 4 Line 10: RHS: $(dq_{air}/dt)_{obs}$ minus simulated terms?

Page 7 Line 7: remove the parenthesis.

Page 9 Line 7: remove the parenthesis.

Page 17: what are the color shadings and streamlines in Figure 1d?

---

## Referee Comment (RC2) · Anonymous Referee #2 · 19 Jul 2018

Title: A Hydrological Cycle Model for the Globally Resolved Energy Balance Model (GREB) v1.0 Authors: Christian Stassen et al. Recommendation: major revision

Summary

The authors present a new version of the Globally Resolved Energy Balance (GREB) model, which adds parameterizations for the three elements of the global hydrological cycle: precipitation, evaporation, and moisture transport. This new version of GREB is more successful in reproducing the behavior of the hydrological cycle in observations and in CMIP5 models, as the authors illustrate with case studies that include interannual variability and greenhouse gas forcing.

The model development and its performance are well presented and sufficiently doc-

umented with figures. I believe, however, that the authors need to explain better how this model can be used. While I realize that this is a technical paper describing the model, I think there needs to be some justification as to why we need this model in the first place.

Major comments

1) It is nice that the new version of GREB is more successful in reproducing certain aspects of the hydrological cycle. On the other hand, given that the new version has more fitting parameters, is this really surprising? Using more parameters gives you a better fit but also carries the risk of overfitting. In particular, the model might be too constrained by present day climate to be useful for climate change projection because basic features of the present climate, such as the width of the Hadley circulation or the position of the ITCZ, may change.

2) Another way the model could be used is for understanding the climate change response of more complex models. For this purpose, it would seem that GREB's mixture of basic principles (e.g. energy balance), ad-hoc parameterization (e.g. standard deviation of omega), and fitting to observations (e.g. mean omega) does not lend itself to interpretation any more than model output itself.

3) section 3.3 What causes f to be 2.5 rather than 1.0? Could there be an error in the calculation? Is this mismatch horizontally uniform?

---

## Referee Comment (RC3) · Anonymous Referee #2 · 19 Jul 2018

Minor comments

1) p. 1, ll. 25-26: I think the authors mean AR5 here (whose models are quite dated by now). Also, "the best possible" is certainly debatable. Perhaps it suffices to say that they are very complex models.

2) Equation (12): Please explain all the variables directly after introducing the equation. In particular, what is u_star? It seems that it should be near-surface wind speed, but then, on p.7 ll. 2-3 it says that wind speed is lower over land for a given u_star, suggesting that it is something else.

3) p. 8, l. 26: Is the good match with observations for Ireland very meaningful if what you are really interested in is the global quality of the data?

4) The English could use a little editing (e.g. number agreement). Some examples: p. 2, l. 19: "parameterisations . . . is described" -> "are described" p. 3, l. 16: "wind and cloud cover field are" -> "fields" p. 3, l. 23: "a autoregressive" -> "an autoregressive" p. 5, l. 13: "precipitation and its seasonal cycle is shown" -> "are shown" p. 5, l. 3: "It, however, has. . ." -> "It has, however, . . ." p. 6, l. 3: "range of uncertainty CMIP5 modelled" -> "of CMIP5"

---

## Author Comment (AC1) · 5 Oct 2018

Dear Min-Hui Lo,

This is the response to the referee comments (RCs) on the manuscript "A Hydrological Cycle Model for the Globally Resolved Energy Balance Model (GREB) v1.0" submitted to GMD by Stassen et al. 2018 (gmd-2018-131). We like to thank the editor and the two anonymous referees for the time and effort spent on reviewing this manuscript and for the many helpful comments they provided. We think the referee comments have helped to substantially improve this manuscript. Please find a point-by-point response to all referee comments. All page and line numbers refer to the original manuscript and might not match up with the revised manuscript. We hope this settles all of the referee concerns and we would be happy to submit the revised manuscript.

Kind regards,
Christian Stassen

**Comments**

*1. Can the authors elaborate the difference between the GREB model and CMIP5 CGCMs? As atmospheric and oceanic circulations are not simulated in the GREB model, it is probable that the GREB simulations give rise to results lacking of dynamical contribution. Does this lacking component play some roles in affecting the performance in the new model?*

Response: CMIP models are earth system models containing several sub-models (i.e. atmosphere, ocean, vegetation, etc.). The atmospheric model would solve the dynamical equations of the atmosphere (i.e. Navier-Stokes). Those lead to internal variability (i.e. weather). GREB is an energy balance model. It does not contain weather or has any internal variability. Thus, reaching equilibrium (i.e. constant temperature) quickly and then remaining at equilibrium if not disturbed by external forcing.

Atmospheric circulation is not dynamically responding but prescribed as an external boundary condition. The sensitivity experiments ENSO & Climate Change in GREB take into account a change in circulation. This is done by adding an anomaly of horizontal winds and omega (and surface temperature) on top of the GREB climatological field. The anomaly is obtained from a composite of El Ninos (or La Ninas) for the ENSO case and the ensemble mean of the CMIP5 RCP8.5 response for the climate change case.

Oceanic circulations are not considered in GREB. The effect of ocean currents on the atmosphere would be reflected through a change in sea surface temperature (SST) which, is part of the GREB models external forcing. Thus, the GREB model would respond to a change in ocean currents through the change of SST.

We changed page 3 line 16-18 to: 'Thus, the GREB model is conceptually very different from the CGCM simulations in CMIP5, as atmospheric and the oceanic circulations are not simulated but prescribed as an external boundary condition in the model. The effect of ocean circulation on the atmosphere is represented only through the sea surface temperature.'
* * *
*2. Due to the fact that CMIP5 CGCMs have biases in simulating circulations (e.g., Yang et al., 2018, Journal of Climate), the differences, at least for circulations, between the GREB model and CMIP5 models could be traced to the differences between prescribed wind fields in ERA-Interim reanalysis product and simulated wind fields. In other words, different background wind states may be part of the reason generating the discrepancy. What if comparing results using the CMIP5 simulated mean fields to force the old/new GREB model? Do authors have insights toward this point?*

Response: We performed a set of experiments where the GREB model is forced with boundary conditions (i.e. horizontal winds and omega) from CMIP5 models. We compared the GREB precipitation anomaly (GREB-ERAInterim forced *minus* GREB-CMIP forced) to the CMIP precipitation anomaly (CMIP *minus* ERAInterim). This showed a high pattern correlation (~0.8). For the majority of the CMIP models we looked at, most of the correlation was caused by forcing

GREB with omega. However, this needs more research and will be done in future work before we can confidently address this.

We added the following to page 12 line 11: 'A very recent study by (Yang et al., 2018) links circulation biases in CMIP models to biases in precipitation and moisture. Forcing GREB with the circulation of CMIP models could shed more light how discrepancies in circulation between CMIP models effect the hydrological cycle in the GREB model.'

*3. If no daily weather systems are simulated, does that mean the temporal integration is performed in the time step of one month or more in the GREB model? I am confused because the model time step is 12hrs as mentioned in Page 3 Line 14. In addition, having no weather system simulated does not mean no internal variability generated in the model. I suggest authors rephrasing this pragraph or providing further explanation.*

Response: There are two different time steps in the GREB model:
- The physics and the tendency equations (i.e. tendencies for the hydrological cycle or surface temperature) are integrated on a 12 hours time step.
- Circulation (Advection and Diffusion) is integrated using a sub-stepping of 0.5 hours or 24 sub-steps. This is necessary for the model to be numerically stable.

We rephrased Page 3 Line 13-16 to: "The tendency equations of the model (i.e. tendency equation of specific humidity) are solved with a time step of 12 hours. For circulation, a shorter time step of 0.5 hours is used. This is necessary for the model to remain numerically stable. The daily cycle of incoming solar radiation is not resolved instead the 24hrs mean incoming solar radiation is used."

If the GREB model has no external forcings it will reach equilibrium quickly, depending on the magnitude of the forcing. After reaching equilibrium the model is stable and the tendencies of the model go towards zero. This means the GREB model is not oscillating around an equilibrium point.

We rephrased Page 3 Line 19-21 to: "Additionally, the GREB model has no internal variability. This means the model will converge to its equilibrium point and the tendency equations converge to zero."

*4. Page 1 Line 8: Does "the hydrological cycle" refer to the hydrological model? Similar mixture appears throughout the abstract.*

Response: Yes. The terms 'hydrological cycle' and hydrological model' are used interchangeably. We changed this in page 1 line 8 and through the manuscript to be more precise.

*5. Page 1 Line 9: The authors should clarify the meaning of "zero order". Does the "order" means numerical convergence rate in time or space?*

Response: With zero order we mean that is was a first guess or a rudimentary approach. The term is clarified in a bit more detail on page line 12: 'The hydrological cycle in the GREB model was

only needed as a zero order estimate to model the latent heat in the energy balance and the atmospheric water vapour levels.' We changed the word on page 1 Line 9 to 'rudimentary' to not cause confusion with order of numerical convergence or order of accuracy.

*6. Page 2 Line 6: Authors may consider mentioning the computational efficiency of idealized model here.*

Response: We changed Page 2 Line 6-7 to: 'Because of their simplicity, they help to develop hypotheses about the processes involved and they can be run fast. The GREB numerical code computes one model year in a few seconds and on a standard personal computer. It therefore is a relatively fast tool, which allows conducting sensitivity studies to external forcing within minutes to hours (Dommenget & Floter, 2011)'

*7. Page 2 Line 23: I suggest re-organizing the paragraphs that describe the GREB model. For example, the modl layer configuration and resolution in Page 3 Line 10 can be introduced before the description of the NCEP climatological fields used in the original GREB model. This may make introduction of the GREB model framework smoother.*

Response: We moved the paragraph describing the GREB model to the beginning of chapter 2

*8. Page 2 Lines 25-26: Any specific reason generating topography from an atmospheric model? Why not using ETOPO dataset?*

Response: We adopted the approach form the original GREB model.

*9. Page 2 Line 29: I suggest providing brief explanation for the reason of changing dataset here and directing further details to section 3.4. Also, the NCEP reanalysis datasets are used during 1950-2008, whereas the ERA-Interim reanalysis data during 1979-2015. Therefore, long-term mean climatology values may be different. What is the results using the NCEP data during 1979-2015?*

[Figure]

Response: The difference between the mean climatologies between NCEP 1950-2008 vs. NCEP 1979-2015 is small compared to difference between NCEP and ERA-Interim (see plot above). We added the following to page 2 Line 28: 'ERA-Interim reanalysis has a higher accuracy than NCEP and a better agreement with observations (Liu et al., 2017)' and page 8 line 28: 'The effect of changing the mean climatology from the years 1950-2008 to 1979-2015 is small compared to the differences between NCEP and ERA-Interim.'
* * *
*10. Page 4 Line 3: How important is the diffusion term compared to advection? Could we ignore diffusion effect in large-scale circulation dynamics and/or thermodynamics?*

Response: The diffusion term is about 1/5 of the magnitude of the advection term (annually and

[Figure]

Figure 1: Annual mean advection (left) and diffusion (right) for the GREB model in kg/m2/s.

globally averaged, see Figure 1 below). Therefore, we did not ignore diffusion. We added: 'The diffusion term is one fifth of the magnitude of the advection term in global average but it is more important in some locations and therefore not ignored in the GREB model (not shown).'
* * *
*11. Page 10 Line 6: ENSO events include both El Niño and La Niña. Is the analysis for La Niña shown in the manuscript? Figure 10 only shows analysis for El Niño composite, but not La Niña. How does response to La Niña look like?*

Response: The improvement in the response of the GREB model to La Niña is similar to the improvement in GREB simulated El Niño. It is however, not shown in the manuscript. We added to Page 10 Line 6: 'La Nina events are qualitatively the same, but with opposite signs (not shown).'
* * *
*12. Page 10 Line 9: Figure 10g shows the improvement in precipitation anomalies. It could be informative also discussing extratropical precipitation response because ENSO-mid-latitude linkages were also well documented. In Figure 10g, no precipitation increases in the Southern U.S. region. Does that indicate ENSO teleconnection is still not resolved well in the new model?*

Response: We focused here on a first order approximation of the response and do not want to discuss all details. The figure is in mm/day and therefore is focussed on the tropics where the absolute response in precipitation is strong. The extratropical response in GREB roughly matches

observations but is weaker than observations. We added the following to page 10 line 11:'… precipitation response is somewhat weaker than observed, especially in the extra tropics.'
* * *
*13. Page 1 Line 15: It is better to mention the full name of "CMIP" in the abstract.*

Response: Changed Page 1 Line 15 to: 'The new hydrological cycle is evaluated against the Coupled Model Inter-comparison Project phase 5 (CMIP5) model simulations, …'

*14. Page 1 Line 16: El Nino -> El Niño (and through the manuscript)*

Response: Changed El Nino to El Niño throughout the manuscript. Changed La Nina to La Niña throughout the manuscript.

*15. Page 1 Line 17: Add the full name of "CGCM".*

Response: Added Coupled General Circulation Models (CGCMs) to Page 1 Line 17.

*16. Page 1 Line 25: What is "CGCMs" stands for? If it refers to "General Circulation Models" at the end of Line 24, it should be abbreviated as GCMs.*

Response: Changed in manuscript.

*17. Page 1 Line 26: "(AR4)" only appears once here, there's no need to provide abbreviation.*

Response: Removed AR4

*18. Page 3 Line 5: Figure 2c and 3c -> Figures 2c and 3c. I found this kind of error appears through the manuscript (e.g., Page 4 Line 4 and Page 5 Line 23). Please check carefully and revise them consistently.*

Response: Changed to Figures 2c and 3c on page 3 line 5 and through the manuscript.

*19. Page 3 Line 7: rcp85 -> RCP8.5 (to be consistent to that in the caption of Figure 11, Page 27).*

Response: Changed in manuscript.

*20. Page 4 Line 10: RHS: (dqair/dt)obs minus simulated terms?*

Response: That is correct! Changed the order in the manuscript

*21. Page 7 Line 7: remove the parenthesis.*

Response: Removed parenthesis

*22. Page 9 Line 7: remove the parenthesis.*

Response: Removed parenthesis and added 'and' between citations.

*23. Page 17: what are the color shadings and streamlines in Figure 1d?*

Response: Added '…850 hPa wind direction (streamline) and strength (shading)' to the caption.

**Referee 2**

**Comments**

*1. I believe, however, that the authors need to explain better how this model can be used. While I realize that this is a technical paper describing the model, I think there needs to be some justification as to why we need this model in the first place.*

Response: We plan to apply the model for studying biases in CMIP models. This could be done by replacing boundary conditions through CMIP boundary conditions. We revised the manuscript to better highlight the use of the GREB model. For example we added the following to page 12 line 11 (also to address RC1.2): 'A very recent study by Yang et al., 2018 links circulation biases in CMIP models to biases in precipitation and moisture. Forcing GREB with the circulation of CMIP models could shed more light how discrepancies in circulation between CMIP models effect the hydrological cycle in the GREB model.'

*2. It is nice that the new version of GREB is more successful in reproducing certain aspects of the hydrological cycle. On the other hand, given that the new version has more fitting parameters, is this really surprising? Using more parameters gives you a better fit but also carries the risk of overfitting. In particular, the model might be too constrained by present day climate to be useful for climate change projection because basic features of the present climate, such as the width of the Hadley circulation or the position of the ITCZ, may change.*

Response: We addressed the problem of overfitting by different approaches: first we tested the development of the model in step-wise building up the complexity (see section 3). Secondly, we did a number of response experiments that test the model's skill beyond the information used to fit the parameters. For this we did three tests: Seasonal cycle, El Nino and climate change. In all three the new model showed skills in simulating changes in the hydrological cycle that would not have been achieved by overfitting the model.
We added some additional information in the introduction of Section 4 to better highlight this problem.

*3. Another way the model could be used is for understanding the climate change response of more complex models. For this purpose, it would seem that GREB's mixture of basic principles (e.g. energy balance), ad-hoc parameterization (e.g. standard deviation of omega), and fitting to observations (e.g. mean omega) does not lend itself to interpretation any more than model output itself.*

Response: We agree that the mixture of basic principles and ad-hoc parameterisations helps understanding the climate change response of more complex models (i.e. their biases). The boundary conditions of GREB could for example be replaced with climatologies of CMIP models (i.e. replacing horizontal winds from ERA-Interim with horizontal winds from one CMIP model). By replacing only one or all boundary conditions would help to gain insight where changes in RCP-scenarios come from or where biases in the hydrological cycle originate from. This is indeed what we think could be a useful application of this GREB model.

We added the following to page 12 line 11 (also to address RC1.2 & RC2.0): 'A very recent study by (Yang et al., 2018) links circulation biases in CMIP models to biases in precipitation and moisture. Forcing GREB with the circulation of CMIP models could shed more light how discrepancies in circulation between CMIP models effect the hydrological cycle in the GREB model.'

*3. section 3.3 What causes f to be 2.5 rather than 1.0? Could there be an error in the calculation? Is this mismatch horizontally uniform?*

Response: There are several sources of uncertainties:
- The value of the scaling height we use in GREB is larger than literature values
- The fact that GREB is a single layer model
- The coarse resolution of the GREB horizontal grid
- A mismatch of the omega climatology
- Calculating circulation as residual

f is a constant fitting parameter it has no dimensions (see table 2 page 15). It could be fitted to different regions (i.e. tropics only or extra-tropics only) to get estimate if it is horizontally uniform.
We added to page 8 line 14: '… vertical velocities may not perfectly match because of the coarse resolution, GREB uses a scaling height of water vapour that is larger than literature values and calculating circulation as residual could contain other uncertainties.

*4. p. 1, ll. 25-26: I think the authors mean AR5 here (whose models are quite dated by now). Also, "the best possible" is certainly debatable. Perhaps it suffices to say that they are very complex models.*

Response: Changed line 25-26 to: 'CGCMs evaluated by the Intergovernmental Panel on Climate Change (IPCC) for the fifth assessment report (AR5), are among the most complex simulations of the climate system.'

*5. Equation (12): Please explain all the variables directly after introducing the equation. In particular, what is u_star? It seems that it should be near-surface wind speed, but then, on p.7 ll. 2-3 it says that wind speed is lower over land for a given u_star, suggesting that it is something else.*

Response: U_star is the absolute wind climatology explained in Table 2. We introduced Table 2 before any equation is mentioned in section 2.

*6. p. 8, l. 26: Is the good match with observations for Ireland very meaningful if what you are really interested in is the global quality of the data?*

Response: Deleted: 'and Mooney et al. (2011) found a higher correlation of surface temperature in ERA-Interim to observations then NCEP in Ireland'
* * *
*7. The English could use a little editing (e.g. number agreement). Some examples: p. 2, l. 19: "parameterisations . . . is described" -> "are described" p. 3, l. 16: "wind and cloud cover field are" -> "fields" p. 3, l. 23: "a autoregressive" -> "an autoregressive" p. 5, l. 13: "precipitation and its seasonal cycle is shown" -> "are shown" p. 5, l. 3: "It, however, has. . ." -> "It has, however, . . ." p. 6, l. 3: "range of uncertainty CMIP5 modelled" -> "of CMIP5"*

Response: Thank you for pointing this out. We carefully read over the manuscript and changed all errors pointed out above plus those we additionally found throughout the manuscript.

---

## Author Response (AR2)

Dear Min-Hui Lo,

This is the response to the referee reports on the revised manuscript "A Hydrological Cycle Model for the Globally Resolved Energy Balance Model (GREB) v1.0" submitted to GMD by Stassen et al. 2018 (gmd-2018-131). We like to thank the editor and the two anonymous referees for the time and effort spent on reviewing this revised manuscript and for the additional feedback and comments they provided. We think the referee comments have helped, again, to improve this manuscript. If we understand it correctly report #3 is from a third anonymous referee we therefore updated the acknowledgments to thank all three referees for their time.
Please find a point-by-point response to all referee comments. All page and line numbers refer to the latest manuscript. We hope this settles all of the referee concerns and our paper is ready for publication.

Kind regards,
Christian Stassen, on behalf of all authors

**Referee report #1**

*I am satisfied with the aurthors' point-by-point replies mostly. I am still curious about the El Niño and La Niña forcings. The authors replied that the GREB model responses to La Niña are similar those to El Niño. However, the SST anomaly patterns of La Niña and El Niño are asymmetric in general. How do different SST anomaly patterns lead to similar GREB model responses?*

Response: The response pattern in GREB to La Nina is different to the response pattern to El Nino. We meant that the skill of GREB to simulate El Nino is similar to simulating La Nina. Please see below for a figure of the GREB response to La Nina.
We changed the following on page 14 line 12: 'The skill of simulating La Niña events are qualitatively the same.'

[Figure]

*Figure 1: The La Niña response of the hydrological cycle in: observations for precipitation (a) in mm/day, evaporation (b) and circulation (c) in kg/m2/s (upper), original GREB model for precipitation (d), evaporation (e) and circulation (f) (middle) and the improved GREB model for precipitation (g), evaporation (h) and circulation (i) (lower). GREB uses prescribed anomalies from a La Niña composite mean of surface temperature, horizontal winds and vertical winds (omega).*

**Referee report #2**

*Hydrological cycle is a loaded term. Usually, it refers to water cycling between land and ocean and atmospheric reservoirs. Thus, it has fluxes and storages. Since you only simulate a few selected fluxes (and exclude storage and other fluxes such as runoff), I would suggest adding a short justification on why these three fluxes are important and providing an acknowledgement and justification for not implementing other fluxes and storage terms (perhaps in order to maintain the speed and simplicity of the model). Due to this, I suggest replacing 'hydrological cycle' with 'hydrology variables' wherever possible, and 'hydrological cycle model' to 'hydrology component' of GREB.*

Response: We understand that the term hydrological cycle refers to more fluxes than the ones simulated by GREB. However, we would like to keep the term 'hydrological cycle' as is to stay consistent with the previous publications of the GREB model and a current paper under review at GMD. We highlight on P3 Ln16-17 that hydrological cycle in GREB means evaporation, precipitation and water vapour transport. We additionally added the following to P3 Ln23-24: 'In addition, wind, cloud cover and soil moisture fields are seasonally prescribed boundary conditions and …' and on page 3 ln 27 and following '…as atmospheric circulations, cloud cover and changes to soil moisture are not simulated but prescribed as external boundary conditions in the model. This leads to some parts of the hydrological cycle not being simulated in the GREB hydrological cycle model (i.e. runoff).'

*P2 Ln 3: Avoid use of 'like' since it is ambiguous and somewhat informal. Suggested alternative: 'such as'.*

Response: We replaced 'like' to 'such as' throughout the manuscript.

*P2 Ln 9: GREB abbreviation appears before the complete name (Ln 11) Suggested minor text flow changes: First introduce GREB model, describe what is unique about it (it is fast and simple), describe its hydrology component and what it is lacking, and the need to upgrade it.*

Response: We revised this section (P2 Ln21 onwards), following the suggestion to first introduce GREB, describe why GREB is unique, describing the hydrology part and the motivation of this paper.

*P2 Ln 17: keep one tense (present) consistent throughout the paper. Avoid using future tense, since you have already finished the study, not proposing to do so.*

Response: We changed the tense to present tense and avoided using future tense throughout the paper.

*P2 Ln 26: Surface -> 'land and ocean surface'*

Response: We changed 'surface' to 'land and ocean surface'.

*Ln 25 onwards on P2: Outputs from GREB are described but not the input. Again, suggest a text flow change: describe GREB, what is unique about it, what are the forcing fields and what are the output fields. Then describe it logistics- time step, datasets used for forcing etc.*

Response: We rephrased this section (P3 Ln20 onwards) for a better text flow.

*P3 Ln3: Write the full name of CMIP5, since this is the first time in the article text that you are referring to it.*

Response: We added the full name of CMIP5.

*P3 Ln 16: At this point in the paper, it is unclear why precipitation observations are used. Please supplement the sentence with a short explanation.*

Response: We rephrased P4 Ln18:' Precipitation from reanalysis products is influenced by the underlying CGCM (Gehne et al., 2016) and is therefore taken from observations from the Global Precipitation Climatology Project (GPCP) (Adler et al., 2003).'

*Figure 3: What specific metric is used to denote the seasonal cycle? Please explain.*

Response: We revised the figure caption of figure 3, figure 7 and P4 Ln23 to specify that we used DJF minus JJA as seasonal cycle.